# Use of *Galleria mellonella* as a Model for Insect Vector Transmission of the Foodborne Pathogen *Campylobacter jejuni* in Broiler Chickens: A Pilot Study

Gemma Chaloner [1] , Lizeth Lacharme-Lora [2], Amy Wedley [2] and Paul Wigley [2,3,*]

1 School of Life Sciences, University of Liverpool, Liverpool L69 3BX, UK
2 Department of Infection Biology and Microbiomes, Institute of Infection, Veterinary and Ecological Sciences, Leahurst Campus, University of Liverpool, Neston CH64 7TE, UK
3 Bristol Veterinary School, Langford Campus, University of Bristol, Bristol BS40 5DU, UK
* Correspondence: paul.wigley@bristol.ac.uk

**Abstract:** There is growing pressure to find a way to eradicate or reduce the levels of foodborne pathogens such as *Campylobacter* in broiler chickens, whilst limiting the use of antimicrobials. For *Campylobacter*, there is currently no vaccine and on-farm biosecurity alone is insufficient to prevent colonization of broiler chicken flocks. Dipteran flies are proven carriers of *Campylobacter* and their entry into broiler houses may contribute to its transmission to broiler chickens. As there is currently no experimental vector transmission model for *Campylobacter* and chickens, we decided to examine experimentally whether *Galleria mellonella* could be used as vector to transmit *Campylobacter* to broiler chickens. More recently, the use of live insect feed has been proposed both for its nutritional qualities and improving bird welfare through the encouragement of natural foraging behaviours and it is unclear any risk this poses in terms of pathogen transmission. In this study, day-old chicks (*n* = 29) were obtained from a commercial hatchery. At three weeks of age, birds were split into 4 This groups; Group 1 was infected via oral gavage with $10^6$ cells of *C. jejuni*-M1, Group 2 was fed *Galleria mellonella* infected with $10^6$ cells of C. jejuni-M1, Group 3 was fed uninfected *Galleria mellonella*, whilst the remaining group was unchallenged. Cloacal swabs were taken at 2, 4, and 6 days post-infection (dpi) to follow transmission and at 8 dpi birds culled and *C. jejuni* load quantified in the caeca and liver. At 8 dpi, all birds in both the *Campylobacter* gavage group and those in the group fed the *Campylobacter* infected *Galleria mellonella* were Campylobacter positive, whereas those fed uninfected *Galleria mellonella* and the control group were all Campylobacter negative. The mean caecal *Campylobacter* load in the *Campylobacter* gavage group was $1.7 \times 10^{10}$ per gram compared with $8.6 \times 10^9$ in the group fed the *Campylobacter*-infected *Galleria mellonella*. No liver positives were found in any of the groups. Our findings indicate that feeding broiler chickens with the vector *Galleria mellonella* infected with *C. jejuni*-M1 is sufficient to establish colonisation with *C. jejuni*. We propose that *Galleria* can be used as an easy and flexible model for vector transmission of foodborne pathogens in chicken.

**Keywords:** insect vector; *Campylobacter*; *Salmonella*; transmission; infection model; broiler chicken

## 1. Introduction

The control of zoonotic bacterial foodborne infection remains a key public health issue for both poultry meat and egg production. Control of *Salmonella enterica* through vaccination and biosecurity has had some success [1], but such controls are either not available or are less effective in the control of *Campylobacter jejuni*. There are multiple sources through which pathogenic bacteria may enter chicken production, including vertical transmission or hatchery infection, contaminated feed, biosecurity breakdowns allowing entry on workers' clothing and boots, and introduction via vermin and wild birds. Transmission via invertebrates largely as fomite vectors has been recognized as a potential source

of foodborne pathogens with the potential for long-term persistence in adult and larval forms [2]. A range of arthropod species including beetles, mealworms and flies have been shown to have the capacity to harbour pathogens [3–5]. The poultry ectoparasite red mite (*Dermanyssus gallinae*) has also been implicated as a vector for systemic *Salmonella* infection [6]. There is strong evidence for Dipteran flies acting as a vector for *C. jejuni* entry into poultry houses and they are likely to play a role in the higher levels of *C. jejuni* in European chicken flocks in summer [7]. Provision of fly screens in housing leads to a decrease in prevalence of *C. jejuni* in broiler flocks [8].

Recently, the use of both insect-derived protein and feeding with live insects has attracted considerable interest. Black soldier fly larvae (*Hermetia illucens*) and mealworms (*Tenebria molitor*) are the main species used and there are claims that there are both nutritional and welfare benefits for their use [9]. However, as both species can carry foodborne pathogens, the role of live insect feed as potential vectors also needs to be considered.

Working with insects in transmission studies is challenging given both their size and the availability of a regular and consistent source of insects. A potential alternative is the use of insects or insect larvae used as live food by reptile keepers which can be readily sourced from pet stores or online suppliers. Larvae of the greater wax moth (*Galleria mellonella*), also called wax worms, have in recent years become widely adopted as alternative infection models including for *Salmonella* and *Campylobacter* [10,11] and as a testing platform for antimicrobials and probiotics. *Galleria* larvae are relatively large, robust and can be readily injected into their haemocoel. In most studies, death or melanisation of larvae are used as endpoints for virulence, but in our previous studies using the challenge of *Galleria* larvae with *C. jejuni* [10], we found that it was possible to infect the larvae with certain strains of bacteria without significant signs of pathology. On this basis, we considered that colonized wax worms could be used experimentally as a model to assess insect vector transmission to chickens in a simple and reproducible manner.

## 2. Materials and Methods

### 2.1. Bacterial Strains and Culture Conditions

*C. jejuni* M1 was grown from stocks maintained at $-80\,^{\circ}\text{C}$ on Columbia blood agar (Lab M, Heywood, Lancashire, UK) supplemented with 5% defibrinated horse blood (Oxoid, Basingstoke, Hampshire, UK) for 48 h in microaerobic conditions (80% $N_2$, 12% $CO_2$, 5% $O_2$, and 3% $H_2$) at $41.5\,^{\circ}\text{C}$. Liquid cultures were grown for 24 h in 10 mL of Mueller–Hinton broth (MHB) in microaerobic conditions at $41.5\,^{\circ}\text{C}$ and adjusted by dilution in fresh MHB to a final concentration of $10^6$ CFU/mL.

### 2.2. Experimental Animals

Twenty-nine Ross 308 broiler chicks were obtained from a local hatchery. Birds were divided into three groups of 7 and a group of 8. Chicks were reared in floor pens in the University of Liverpool biosecure poultry unit and maintained on clean wood shavings at $30\,^{\circ}\text{C}$ and given ad libitum access to water and a pelleted vegetable protein-based diet (SDS, Witham, Essex, UK) as described previously [12]. All animal work was conducted in accordance with UK legislation governing experimental animals under project licence 40/3652 and was approved by the University of Liverpool ethical review process prior to the award of the licence. All animals were checked a minimum of twice daily to ensure their health and welfare.

### 2.3. Infection Experiment

#### 2.3.1. *Campylobacter* Infection (Group1)

At 21 days of age, birds were infected by oral gavage with $2 \times 10^6$ cells of *C. jejuni* M1 suspended in 0.2 mL of MHB as described previously [12].

### 2.3.2. *Galleria* Vector Infection (Group2)

*Galleria mellonella* larvae (waxworms) were obtained from a commercial supplier (Live Foods Direct, Sheffield, UK) and used within 48 h of delivery. Any small or moribund larvae were discarded. *G. mellonella* infection was performed as previously described [11]. Briefly, final instar *G. mellonella* larvae (2–3 cm long weighing 180–250 mg each) were inoculated with $10^6$ CFU/10 μL of an overnight culture of each *C. jejuni* M1 by microinjection into the haemocoel using a Hamilton Syringe (Hamilton, Switzerland). A total of 80 infected larvae (10 per bird) were introduced to the pen of Group 2 and the birds were allowed to forage and consume the larvae. All larvae were consumed within 5 min of introduction.

### 2.3.3. Controls (Group 3)

Birds were mock infected with PBS using the methods described above for Group 1.

### 2.3.4. *Galleria* Controls (Group 4)

*Galleria* were mock infected using the methods described above for Group 2, replacing the *C. jejuni* culture with PBS. The group was fed 70 larvae in total, 10 larvae per bird. However, some birds in the group will have eaten more larvae and some less of the larvae, as the larvae were placed into the pen rather than individually fed to each bird.

At 2, 4 and 6 days post-infection, cloacal swabs were taken from each bird to determine faecal shedding of *C. jejuni* and processed as described below. At 8 days post-challenge (29 days of age), all birds were killed by neck dislocation. At post mortem analysis, caecal contents were taken for bacteriological analysis and liver samples taken to determine any extraintestinal spread of *C. jejuni*. Samples were processed as described below.

### 2.4. Bacteriological Analysis

### 2.4.1. Cloacal Swabs

As a measure of colonisation, determination of faecal shedding of *C. jejuni* at 2, 4 and 6 days post-challenge was carried out using the semi-quantitative approach to enumeration from cloacal swabs [12]. Briefly, cloacal swabs were taken and eluted in 2 mL modified Exeter broth consisting of 1100 mL nutrient broth (Lab M, Heywood, Lancashire, UK), 11 mL lysed defibrinated horse blood (Oxoid, Basingstoke, Hampshire, UK), 10 mL *Campylobacter* Enrichment Supplement SV59 (Mast Diagnostics, Liverpool, UK) and 10 mL *Campylobacter* Growth Supplement SV61 (Mast Diagnostics, Liverpool, UK). Swabs were then plated onto mCCDA agar supplemented with SV59. Enriched swabs were incubated at 41.5 °C for 48 h before re-plating onto mCCDA agar supplemented with SV59. Plates were incubated for 48 h at 41.5 °C under microaerobic conditions before being scored for the level of bacterial growth. Bacterial growth was recorded as growth by direct plating onto mCCDA, or where *Campylobacter* was detected by direct plating, or following enrichment culture.

### 2.4.2. Caecal Load

In order to determine the levels of *C. jejuni* colonization in each of the groups, caecal contents were collected from individual birds at necropsy and diluted in 9 volumes of maximal recovery diluent (MRD). Serial 10-fold dilutions were made of each sample in MRD and trilicate 20-μL spots were plated onto mCCDA agar supplemented with SV59. The plates were incubated under microaerobic conditions at 41.5 °C for 48 h, and *Campylobacter* colonies were enumerated to give CFU/g of caecal contents.

### 2.4.3. Extraintestinal Spread

In order to determine extra-intestinal spread by *C. jejuni*, liver samples were diluted in 4 volumes of MRD. Following homogenisation by stomaching, 100 μL was spread onto the surface of a mCCDA supplemented with SV59. In addition, 200 μL was eluted in 2 mL of Exeter broth for enrichment. The plates and enriched samples were incubated as described above. *Campylobacter* colonies (if present) were enumerated. If no growth was recorded, the

enriched samples were plated on mCCDA supplemented with SV59 and colonies counted following incubation.

### 2.5. Statistical Analysis

Significance in bacterial levels between groups was made via the Kruskall–Wallis Test through the open-source analysis platform 'R' (the R Project, https://www.r-project.org, accessed on 1 January 2022).

### 3. Results

Ingestion of *Campylobacter*-infected *Galleria* led to infection and faecal shedding in broiler chickens. Cloacal swabs (Table 1) showed that initially greater numbers of birds challenged by gavage were shedding at 2 and 4 days post-challenge, though by 6 days, numbers were greater in the *Galleria*-infected group. Neither the mock infected control group nor the mock-infected *Galleria* controls showed any shedding of *C. jejuni*.

**Table 1.** Total number of broiler chickens shedding *C. jejuni* M1, tested by collection of cloacal swabs at 2, 4 and 6 days post-challenge.

| Group | Number of Positive Cloacal Swabs | | |
|---|---|---|---|
| | 2 dpi | 4 dpi | 6 dpi |
| 1—Challenged (*n* = 7) | 4 | 6 | 5 |
| 2—*Galleria* challenged (*n* = 8) | 3 | 4 | 6 |
| 3—Unchallenged (*n* = 7) | 0 | 0 | 0 |
| 4—*Galleria* unchallenged (*n* = 7) | 0 | 0 | 0 |

Both gavage and *Galleria* challenge led to high levels of colonisation of *C. jejuni* in the caeca (Figure 1). No colonisation was found in either control group nor was *C. jejuni* detected in the liver of any group.

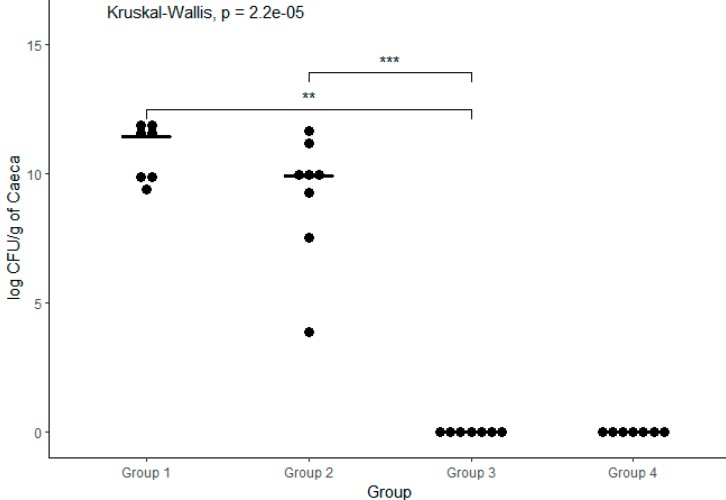

**Figure 1.** Levels of *C. jejuni* M1 in caecal contents of broiler chickens 8 days post-challenge. Group 1 were challenged by oral gavage; Group 2 were challenged by feeding with infected *Galleria mellonella* larvae. Groups 3 and 4 were control groups, given PBS delivered either by *Galleria mellonella* larvae (Group 3) or oral gavage (Group 4). Group sizes were N = 7 for groups 1, 3 and 4 and N = 8 for group 2 (** *p* < 0.05, *** *p* < 0.005).

Although levels of *C. jejuni* caecal colonisation were generally higher in the gavage-challenge group, levels varied from bird to bird and there was no overall statistical difference between the challenged groups (*p* = 0.92). Both challenge groups showed highly significant levels of colonization (*p* > 0.005).

## 4. Discussion

Here we show that *Galleria* larvae represent a simple and effective model to study transmission of foodborne pathogens to chickens. It is clear from our data that *C. jejuni*-infected larvae, when ingested, lead to colonisation of broiler chickens. Based on cloacal swabs, the infection process is slower than direct challenge by gavage, but nevertheless, faecal shedding could be detected in some birds within two days. This slight delay is likely to reflect digestion of the *Galleria* larvae and release of *C. jeuni* that then colonise the chicken intestinal tract.

Whilst this model has limitations, it is inexpensive and relatively simple to deliver. It has potential utility in understanding how transmissible a pathogen, bacterial virulence factors that allow survival in or on an invertebrate host and onward transmission are likely to be via this route. The *C. jejuni* isolate used in this study is highly transmissible, as we have previously shown [13], and it is likely that other isolates and less transmissible pathogens will show differential rates of transmission or may fail to colonise the chicken gut from infected insects. Such information is important in understanding the risk of transmission into housed flocks and mitigations needed to reduce such risk. Arthropod vectors offer challenges to maintaining biosecurity in broiler flocks. The high numbers of insects in summer, added to the fact that *C. jejuni* is more ubiquitous in the natural environment than previously thought [14], and the difficulty of preventing insect ingression on farms all contribute to the risk of this transmission route. Whilst fly screens added to houses can help control transmission via flies such as *Diptera* spp. [8], transmission via other species including beetles and crawling insects cannot be ruled out.

As foraging for invertebrates and small vertebrates is considered part of normal chicken behaviour and diet, the addition of insect protein or live insect larvae, mainly black soldier fly, has nutritional benefits [9]. When using live insect feed, there is the added benefit of encouraging normal behaviour. However, there is a clear risk that any bacterial contamination of insect larvae during rearing could lead to transmission to chickens. Such risk would be lower with processed insect-derived protein that could be dried or heat-treated to remove contamination but is a clear risk from live-fed insects. As such, we suggest caution in the use of live feeds and that steps be made to ensure that live feed production is clear of pathogenic bacteria.

In conclusion, we demonstrate here that *C. jejuni* can be readily transmitted to chickens via an insect vector in an experimental system. Whilst the use of *Galleria* has some limitations, it is a cheap, tractable and effective tool to study insect vector transmission of foodborne pathogens in chicken.

**Author Contributions:** P.W. devised the research concept. All authors planned the research. A.W., L.L.-L. and G.C. conducted the experimental work. P.W., A.W. and G.C. wrote and edited the manuscript. All authors have read and agreed to the published version of the manuscript.

**Funding:** This research received no external funding.

**Institutional Review Board Statement:** All animal work was conducted in accordance with UK legislation governing experimental animals, the Animals (Scientific Procedures) Act 1986, under project licence 40/3652, and was approved by the University of Liverpool ethical review process prior to the submission to the UK Home Office for consideration and approval.

**Informed Consent Statement:** Not applicable.

**Data Availability Statement:** All raw data are available on request.

**Acknowledgments:** We thank Sue Jopson for her technical support.

**Conflicts of Interest:** The authors declare no conflict of interest.

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
