# Peer review of "Use of Galleria mellonella as a Model for Insect Vector Transmission of the Foodborne Pathogen Campylobacter jejuni in Broiler Chickens: A Pilot Study"

_poultry, doi:10.3390/poultry2010005_

Round 1

Reviewer 1 Report

General Comments:

              This manuscript is well written and extremely well designed to answer the question presented by the authors. The topic is definitely of interest to not only the poultry industry, but also the feed industry and other animal husbandry fields. It is regretful that this reviewer must recommend that this manuscript be rejected due to the extremely small sample size and the lack of replications. If the authors would be able to conduct two or three more replications of this experiment with similar results, this reviewer would be delighted to recommend acceptance of the manuscript. General comments listed below are provided to assist the authors in their endeavor.

1)      The sample size is extremely small but adequate for animal experiments with the current emphasis on limiting the number of animals used, however without subsequent successful replications of the results presented the recommendation for rejection must be given.

2)      The materials and methods section needs more detail in regard to the animal husbandry. What bedding was used? Shavings, sand, peanut hulls, rice hulls, etc.? Fresh or used bedding? How densely packed were the animals? 3 birds per m2, etc.? Did the birds have access to fecal and/or cecal droppings of other chicks in the trial?

3)      How were the waxworms maintained? Purchased and used within 24 h? maintained in a colony until used? If maintained in a colony, what were the parameters?

4)      Realizing that this is a draft of a manuscript, this reviewer would like to see the proper use of italics for the genus and species names of bacteria as appropriate. Additionally, the same can be said for the use of superscripts when presenting exponential numbers in proper scientific notation. This needs to be corrected throughout the manuscript.

5)      Why was there a difference in the number of larvae between Groups 2 and 4? 80 infected larvae for Group 2 versus 70 for Group 4. The number of birds in each group were the same. The bias of an additional 1.43 larvae per bird in Group 2 should be extremely small, however it might be considered a confounding factor in some statistical models.

6)      Please provide a reference for the semi-quantitative method of enumeration of cloacal swabs used in this approach.

7)      Statistical methods and programs should be listed in the Materials and Methods section in order to provide a clearer understanding of the methods used for determining significance.

8)      This reviewer was unable to find a citation in the manuscript for reference 12 listed in the References. Please remove the reference or properly cite in the manuscript.

 Specific Comments:

              Pg 3, Line 98 – Delete “each”

              Pg 5, Ln 181 – Change “at” to “in”.

Author Response

We thank the reviewer for their comments.  Several are issues of the publishers re-formatting the original submission and creating extra work for reviewers and authors. Specific points:

1)      The sample size is extremely small but adequate for animal experiments with the current emphasis on limiting the number of animals used, however without subsequent successful replications of the results presented the recommendation for rejection must be given.

The sample size is adequate-power calculations would show 7 per group is fine. Should this be repeated?  I certainly would not ask for this as a reviewer or on animal welfare committees given:

A. There are little variations in the data to controls-it's not chance birds become infected -there is no other source of Campylobacter in such a bio secure environment. 

B. A repeat is unlikely to show any difference but doubles the animal useage. Using the correct number to give significance in the experimental systems avoids repeat. This is good design and meets what would be expected of 3Rs principles and ARRIVE guidelines

2)      The materials and methods section needs more detail in regard to the animal husbandry. What bedding was used? Shavings, sand, peanut hulls, rice hulls, etc.? Fresh or used bedding? How densely packed were the animals? 3 birds per m2, etc.? Did the birds have access to fecal and/or cecal droppings of other chicks in the trial?

Reference added for further husbandry details- we have added clean shavings for clarity. Birds in each group have full access to each other and faecal and caecal droppings. Now this may lead to birds infecting each other, though given the speed of colonisation this is unlikely here and arguably not relevant whether a flock gets infection directly from an arthropod, or indirectly via another bird

3)      How were the waxworms maintained? Purchased and used within 24 h? maintained in a colony until used? If maintained in a colony, what were the parameters?

Purchased as detailed and used within 48 hours of purchase for all our experiments using Galleria. The istar stage used is stable for several weeks until pupation and so this is the model used due to its ready availability as a live pet food. We have added some detail here for clarity.

4)      Realizing that this is a draft of a manuscript, this reviewer would like to see the proper use of italics for the genus and species names of bacteria as appropriate. Additionally, the same can be said for the use of superscripts when presenting exponential numbers in proper scientific notation. This needs to be corrected throughout the manuscript.

The authors also wished the publisher had not changed the format to lose italics for binomials of bacteria and arthropods and messed up the superscripts for bacterial counts etc. These were correct (we have checked) when submitted.

5)      Why was there a difference in the number of larvae between Groups 2 and 4? 80 infected larvae for Group 2 versus 70 for Group 4. The number of birds in each group were the same. The bias of an additional 1.43 larvae per bird in Group 2 should be extremely small, however it might be considered a confounding factor in some statistical models.

Group sizes were 8 v 7 birds. Number of larvae per bird was kept the same. (we had a spare delivered)

6)      Please provide a reference for the semi-quantitative method of enumeration of cloacal swabs used in this approach.

Added  here for clarity. The reference was already cited

7)      Statistical methods and programs should be listed in the Materials and Methods section in order to provide a clearer understanding of the methods used for determining significance.

Yes-this was not sufficiently clear and has been added.

8)      This reviewer was unable to find a citation in the manuscript for reference 12 listed in the References. Please remove the reference or properly cite in the manuscript

This has been lost as part of the formatting errors introduced. this has been addressed.

Reviewer 2 Report

The authors have done a good job to show that the infection of campylobacter could be caused by feeding broiler chickens with the Galleria mellonella larva contaminated with Campylobacter jejuni (C. jejuni). In Page 3, Line 97, “Galleria mellonella larva (waxworms) were inoculated with 106CFU/10μl of an overnight culture of each C. jejuni M1 by microinjection into the haemocoel using a Hamilton Syringe. “ The purpose of this study is trying to demonstrate a model system for evaluating the vector transmission of foodborne pathogens in broiler chickens. However, the carry of C. jejuni in waxworms was artificially injected, instead of naturally took in. By this way, it is hard to demonstrate that the C. jejuni can naturally infect waxworms to become a vector to carry this pathogen to infect broiler chickens. It is more like to inject C. jejuni into capsules, and directly feed all the capsules to broiler chickens. Eventually, the broiler chickens probably will be infected and show clinical signs. However, we are not able to state that the capsule is a vector transmission of        C. jejuni. The authors might prove that waxworms can cause a mechanical transmission, but not prove a biological transmission. 

Author Response

We thank the reviewer for the comments. They recognise some limitations in the model-indeed we note in the discussion it has limitations. However:

  • It would be possible to look at uptake in earlier larval stages. However the readily available Instar stage used here does not allow this. Early stage larvae were briefly available to do this (to test potential probiotics), but the company supplying such larvae has gone out of business. Theoretically one could raise their own larvae, but this removes the convenience of why Galleria has become so readily adopted as it is cheap and easily available
  • Uptake via feed would be a more stochastic process and have more variation in number of bacteria uptaken. Such an approach would likely work, but would need a lot of characterisation and larger animal numbers to overcome any variation.
  • The larvae are certainly not like a capsule-they will have many features including an (albeit basic) immune response, nutritional challenges and a resident microbiome (although minimal effect of the in the haemocoel), as such there are clear features which any bacterial strain would need to survive to allow onward transmission

Reviewer 3 Report

Reviewer #1: In this manuscript, the researchers tried to explain about the Use of Galleria mellonella as a model for insect vector transmission of foodborne pathogens in broiler chickens. It is interesting work and can be accepted after revision.

-     The grammar errors should be checked in the whole manuscript.

-       In abstract, the first four lines should be summarized.

 Moderate English changes required throughout the article

-       In introduction, the main objective has been repeated so it should be refined.

-       Some recent and relevant articles may be added as thousands of articles have been published on this topic.

-       Conclusion should be refined as it is not properly written as per results.

Author Response

We thank the reviewer for their comments.

  • I am unsure where these thousands of articles are published, as beyond the use of dipteran flies on a paper in Scientific Reports  we co-authored there is little to no experimental work on arthropod transmission
  • Whilst the reviewer clearly does not like the modern writing style, other than a few typographical errors and the formatting errors introduced by the journal, we find it hard to find grammatical errors-especially since current versions of MS Word flag these up.

Reviewer 4 Report

In this manuscript, the authors experimentally examine whether Galleria mellonella can be used as a vector for the transmission of Campylobacter to broiler chickens. They show that feeding broiler chickens with the vector Galleria mellonella infected with C. jejuni is sufficient to establish colonisation with Campylobacter. However, the main problem in the article is that the title overstates the role of Galleria mellonella. The authors only examine the effect of Galleria mellonella on Campylobacter infection. In addition, it should be considered in the discussion why C. jejuni was not found in the liver of any of the experimental groups. The abstract contains numerous formatting errors and should be further condensed and refined.

Author Response

We thank the reviewer for the comments which throw up no major scientific issues

  • Title has been changed to reflect comment
  • Liver infection is relatively infrequent in C. jejuni infection and its presence, rather than absence is more notable. Whilst one may expect liver infection with Salmonella, this is perfectly standard for this experimental infection and needs no comment
  • Formatting errors were introduced by the journal in conversion from the original Word file. These have been addressed

Reviewer 5 Report

The present study proposes that Galleria can be used as an easy model for vector transmission of foodborne pathogens in the chicken. This proposed work is interesting and some previous studies already explored this topic. Hereby, very critical and major comments to consider the manuscript for publication:

1. Introduction should be extended to include detailed information about: campylobacteriois as a disease, its clinical impact, epidemiological pattern and zoonotic potential.

2. Research question should be included at the end of the introduction section

3. Brief description of the experimental infection with Campylobacter should be included.

4. Please do consider including a table or figure summarizing the experimental groups, the experimental protocol, dose of infection and references in your work for easier readership.

5. What about molecular identification of the isolates/colonies? You did not confirm your finings and your work is just based on morphology of the colonies?

6. What about histopathological examination of the target organs? At least in case of liver which did show any positives? Demonstration of the histopathological lesions is very important.

7. Discussion is almost empty. You should extend your discussion using your findings and comparing it’s with what was reported in the previous work. For example, what about the explanation of no liver positive colonies were obtained?

8. I suggested that the manuscript should be short communication due to the humble presented findings and methodology.

9. Conclusion should be elaborated and preferably in a separate section.

10. The subsections after conclusions (supplementary material, data

availability statement, etc) MUST be modified according to journal

instructions or omitted if not available. Authors left them as they are in the

word template and it is a bit strange to leave the template as it is.

11. The Approval date of the animal experiment should be included in institutional review broad statement at the end of the manuscript.

Given the above comments, my suggestion is major revision and manuscript can be considered after addressing my comments.

Author Response

This has review is from the first version-these comments have been addressed

 by prior revisions

Round 2

Reviewer 1 Report

The authors have made the necessary corrections and have provided their reasoning behind the use of so few animals. However, this reviewer still wishes to object to publication based on the lack of replication. In the event this reviewer is overridden, the following comments are provided to the authors for their use. 

General Comments: 

In the materials and method section, the paragraph on Pg 3, Lines 113-118 should be separated from the previous paragraph to indicate that this was applied to all four treatment groups, not just group 4. 

Please provide a commercial source, reference, etc., for the maximal recovery diluent. 

Please provide a reference for the statement on Pg 5, Ln 188-189 regarding Salmonella being a less transmissible pathogen than Campylobacter. 

Specific Comments: 

1) Pg 2, Ln 65 - remove the word "including"

2) Pg 2, Ln 68 - consider changing the word "using" to another word to ensure that the reader is clear of the results and importance of the previous research.

3) Pg 5, Ln 185 - this sentence appears to be missing a conjunction and as such is unclear to the reviewer.

Author Response

We are not going to agree on the repeats (and your opinion is valid, but so is the counter argument based on 3Rs) but the title is changed to reflect the pilot nature of the work.

Specific points

MRD is not a commercial product, but one we produce to improve Campylobacter recovery over PBS or saline. It has been used by us and others in previous (as referenced) studies.

We have removed specific reference to Salmonella transmissibility-we know this to be the case experimentally, but these data are yet unpublished.

We have addressed the other recommendations as far as we could-we cannot find one  that was suggested

We thank the reviewer for their time